# The Distinct Assignments for Hsp90α and Hsp90β: More Than Skin Deep

**DOI:** 10.3390/cells12020277

**Published:** 2023-01-11

**Authors:** Cheng Chang, Xin Tang, David T. Woodley, Mei Chen, Wei Li

**Affiliations:** Department of Dermatology and the Norris Comprehensive Cancer Centre, University of Southern California Keck Medical Center, Los Angeles, CA 90033, USA

**Keywords:** heat shock protein-90, Hsp90-alpha, Hsp90-beta, distinctions, Hsp90 inhibitors, therapeutics

## Abstract

For decades, the undisputable definition of the cytosolic Hsp90α and hsp90β proteins being evolutionarily conserved, ATP-driven chaperones has ruled basic research and clinical trials. The results of recent studies, however, have fundamentally challenged this paradigm, not to mention the spectacular failures of the paradigm-based clinical trials in cancer and beyond. We now know that Hsp90α and Hsp90β are both ubiquitously expressed in all cell types but assigned for distinct and irreplaceable functions. Hsp90β is essential during mouse development and Hsp90α only maintains male reproductivity in adult mice. Neither Hsp90β nor Hsp90α could substitute each other under these biological processes. Hsp90β alone maintains cell survival in culture and Hsp90α cannot substitute it. Hsp90α also has extracellular functions under stress and Hsp90β does not. The dramatic difference in the steady-state expression of Hsp90 in different mouse organs is due to the variable expressions of Hsp90α. The lowest expression of Hsp90 is less than 2% and the highest expression of Hsp90 is 9% among non-transformed cell lines. The two linker regions only take up less than 5% of the Hsp90 proteins, but harbor 21% of the total amino acid substitutions, i.e., 40% in comparison to the 86% overall amino acid homology. A full understanding of the distinctions between Hsp90α and Hsp90β could lead to new, safe and effective therapeutics targeting Hsp90 in human disorders such as cancer. This is the first comprehensive review of a comparison between the two cytosolic Hsp90 isoforms.

## 1. Introduction

When we searched PubMed for all articles with “Hsp90” alone, “Hsp90α” alone, “Hsp90β” alone or “Hsp90α and Hsp90β” present in the title, the total numbers of publications were 4237, 171, 96 and 17, respectively (as of 1 December 2022). The numbers of publications using a similar search mechanism with “Heat Shock Protein 90”, “Heat Shock Protein 90alpha”, “Heat Shock Protein 90beta” or “Heat Shock Protein 90alpha and Heat Shock Protein 90beta” present in the titles were 1222, 55, 14 and 0, respectively. The publications of these separate searches did not all overlap. Moreover, it is impossible to reach any consensus from the individual findings of these reports rather than the following general conclusions: (1) Hsp90 proteins act as intracellular chaperones for almost the entire cellular signaling networks; (2) Hsp90α and Hsp90β show both compensatory and non-compensatory duties depending upon cellular contexts; (3) the ATP-binding inhibitors, such as 17-AAG, target all Hsp90 family members and (4) Hsp90α, but not Hsp90β, has extracellular functions [1,2,3,4,5,6,7].

## 2. Cytosolic Hsp90 Isoforms

In vertebrates, two distinct Hsp90 genes encode the cytosolic Hsp90α (HSPCAL4, 14q32.31, MIM# 140571) and Hsp90β (HSPCB, 6p21.1, MIM# 140572), respectively. In addition, two organelle-residing isoforms, Grp94 and TRAP1, also belong to the Hsp90 protein superfamily [8]. As shown in Figure 1A, the human Hsp90α and Hsp90β share approximately 86% amino acid homology and differ at a total of 99 amino acid residues along their full-length 732- (Hsp90α) and 724- (Hsp90β) amino acid sequences. The differences in amino acids include 58 conservative substitutions, 41 non-conservative substitutions and 12 (variable numbers of amino acid) deletions in Hsp90β. The substitutions or deletions are not evenly distributed along the two proteins. It is noticeable that there are 21 amino acid substitutions and three amino acid deletions within the 32 amino acid sequences of the linker region (LR), resulting in a reduced homology between Hsp90α and Hsp90β of 40%. Accordingly, as shown in Figure 1B, the recombinant Hsp90α protein migrates slightly slower than Hsp90β in SDS gel electrophoresis. For unknown reasons, the recovery of recombinant Hsp90β from cultured bacteria, even using various improved strains, could only reach 10–20% of recombinant Hsp90α protein production [9]. In the absence of extracellular stress, the steady-state levels of Hsp90α and Hsp90β appear to be similar in cultured cell lines [10]. However, their levels of expression vary dramatically in different mouse organs [11,12,13]. As shown in Figure 1C, while the expression of Hsp90β remains relatively constant among different mouse organs (Note: it is known that beta-actin is uncharacteristically lower in heart tissue), the variations in Hsp90α expression, lowest in the heart and highest in the testis, have as much as a 20-fold difference. While the overall ATPase activity (note, ATP hydrolysis, instead of ATP/ADP exchange) of the purified Hsp90 proteins is uncharacteristically lower than other ATPases under similar testing conditions, there has been no report on the relative ATPase activity between Hsp90α and Hsp90β. The ATPase activity of purified Hsp90α and Hsp90β proteins may nevertheless prove biologically irrelevant, since the ATPase activity of Hsp90s has been shown to be also regulated by interaction with specific co-chaperones. Moreover, the ATP-dependent client protein activation by Hsp90 has been localized to so-called ‘mature’ complexes [14]. Therefore, the physiological ATPase activity of client protein- and/or co-chaperone-bound Hsp90 inside living cells could be surprisingly more volatile. 

## 3. Studies of Hsp90α and Hsp90β Genes in Lower Organisms

Convincing evidence for the importance of Hsp90 genes is their critical roles during the germline or tissue-specific development of various animal models throughout their evolution. The deletion of the *Escherichia coli* Hsp90 homolog HptG was viable but showed growth disadvantages as the environmental temperature increased [15], indicating the requirement for HptG to deal with environmental stress. However, much of the pioneering investigations into the machinery of Hsp90 was conducted in yeasts. In the budding yeast *Saccharomyces cerevisiae*, cells with homozygous mutations for both HSP82 (Hsp90α) and HSC82 (Hsp90β) die at any temperature, in which higher levels of Hsp90 correlated with a higher tolerance for cell growth [16]. While HSP82 and HSC82 share 97% amino acid identity, purified HSP82 and HSC82 proteins exhibit variable ATPase activities, distinct client interactions and different sensitivities of single isoform-expressing cells in growth in response to stress [17]. In *Schizosaccharomyces pombe*, the Wee1 tyrosine kinase requires Hsp90 (swo1–26) to regulate the cell cycle [18]. In *Drosophila*, mutations in the only known Hsp90-related gene, Hsp83 (*E(sev)3A*), are lethal. The cDNA sequence of *E(sev)3A* shows 72% and 74% similarities to those of human Hsp90α and Hsp90β, respectively. Interestingly, the amino acid sequence identities of the *Drosophila* Hsp83 (717 a.a.) to the human Hsp90α (732 a.a.) and Hsp90β (724 a.a.) are both exactly 78.5%. Cutforth and Rubin showed that the *Drosophila* Hsp90 (*E(sev)3A*) was required for downstream signaling by the tyrosine kinase receptor, the *sevenless* receptor, leading to R7 photoreceptor neuron differentiation [19], apparently via the Raf kinase pathway [20]. More interestingly, Yue and colleague reported a critical function for *Drosophila* Hsp90 (*E(sev)3A*) during spermatogenesis, which appeared to be most sensitive to the loss of *E(sev)3A* [21]. 

## 4. Studies of Hsp90α vs. Hsp90β in Mammalian Cells

A dozen studies compared Hsp90α vs. Hsp90β side-by-side in various cellular functions. Kuo et al. reported that Hsp90β is involved in CpG-B ODN signaling but did not carry out experiments to address the specific role of Hsp90α [22]. Using a similar approach, Bouchier-Hayes and colleagues showed that Hsp90α is a key negative regulator of heat shock-induced caspase-2 activation, suggesting a possible mechanism for Hsp90α involvement in anti-apoptosis [23]. This study did not clarify whether Hsp90α acts alone or still requires the co-participation of Hsp90β, i.e., whether the downregulation of Hsp90β affects the status of caspase-2. Houlihan et al. compared Hsp90α and Hsp90β for MHC class II presentation and reported that the disruption of either Hsp90α or Hsp90β expression inhibited the class II presentation of the exogenous and endogenous GAD Ag [24]. Chatterjee et al. recently showed that Hsp90β plays a more important role than Hsp90α in the control of multiple myeloma cell survival [25]. A very interesting but previously overlooked study by Passarino and colleagues reported a healthy Caucasian human bears a missense mutation in the Hsp90α gene that impairs the translation of the protein product, suggesting that Hsp90α is unessential for life [26]. Taherian et al. compared the binding of Hsp90α and Hsp90β to client proteins and co-chaperones in mammalian cells and Xenopus oocytes and reported that Hsp90α and Hsp90β exhibit similar interactions with co-chaperones, but significantly different behaviors with client proteins under stress conditions, inconsistent with the theory of co-chaperones determining client binding specificity [27]. Cortes-González et al. showed that Hsp90α enhances, whereas Hsp90β reduces, NO and O(2)(–) generation by eNOS via modulating eNOS conformation and the phosphorylation state [28]. The KCNQ4 channel plays a critical role in DFNA2, a subtype of deafness with progressive sensorineural hearing loss. Gao and colleagues compared Hsp90α and Hsp90β for controlling KCNQ4 homeostasis and found that overexpressed Hsp90β could restore KCNQ4 surface expression, although it was insufficient to rescue the function of KCNQ4 channels [29]. When skin is injured and blood vessels clot, the local environment becomes ischemic. Jayaprakash and colleagues demonstrated that Hsp90α and Hsp90β work together to promote cell motility in wounded skin and accelerate wound closure, in which Hsp90β binds to the cytoplasmic tail of the LDL receptor-related protein-1 (LRP-1) and stabilizes the receptor at the cell surface. Hsp90α, however, is secreted by the cell into the extracellular space, where it binds and signals through the LRP-1 receptor to promote cell motility, leading to wound closure [30]. Idiopathic pulmonary fibrosis (IPF) is a progressive lung disease characterized by apoptosis-resistant myofibroblasts and the excessive production of ECMs overtaking the normal lung tissue space. Consistent with the findings of Jaraprakash and colleagues during skin wound healing, Bellaye et al. showed that the intracellular form of HSP90β stabilizes the LRP-1 receptor to amplify the functionality of eHsp90α’s extracellular reparative function [31]. Recently, Zou et al. used CRISPR/Cas9 gene-editing technology to knock out Hsp90α and Hsp90β in MDA-MB-231 breast cancer cells. They found that Hsp90α knockout had little effect on the survival and doubling time of the cancer cells, but specifically nullified the cells’ ability to migrate and invade in the absence of serum support in vitro and form tumors in nude mice. More surprisingly, the lost ability of migration and invasiveness in vitro and tumorigenicity in vivo in the Hsp90α knockout cancer cells could be fully rescued by extracellular supplementation or injection to circulation with human recombinant Hsp90α, but not Hsp90β, protein. In contrast, a similar attempt to knock out Hsp90β failed to obtain any viable cell colonies during drug selection [9]. These authors went on to demonstrate that the major difference that distinguish Hsp90α from Hsp90β with regard to having extracellular functions or not is a dual lysine motif, K270/K277. This motif is only present in Hsp90α and substituted with G262/T269 in Hsp90β, as schematically shown in Figure 2A. Swapping between K270/K277 in Hsp90α and G262/T269 in Hsp90β completely abolishes the extracellular functions of Hsp90α and, in reverse, grants Hsp90β the extracellular functions of Hsp90α. This extracellular function-determining lysine motif is evolutionarily conserved in members of the Hsp90α subfamily (Figure 2B) and is substituted in members of the Hsp90β subfamily (Figure 2C). Similarly, it is of great interest to identify the key amino acid motifs that distinguish the specificity of the intracellular chaperone functions between Hsp90α and Hsp90β, which could serve as the targets for Hsp90α- and Hsp90β-specific inhibitors. 

## 5. Role of Hsp90α and Hsp90β during Mouse Development 

Since 2000, several mouse genetic studies have provided perhaps the strongest evidence that Hsp90α and Hsp90β, albeit an 86% amino acid identity, have distinct and non-compensating functions during mouse development. Voss et al. tried to generate Hsp90β mutant mice by gene trap insertion into the exon 9 and reported that heterozygous Hsp90β mutant mice were normal, whereas the homogenous Hsp90β mutant embryos remained normal by E9.0/9.5 but died a day later due to a developmental defect in the placental labyrinth formation, even in the presence of a normal level of Hsp90α [32]. Grad and colleagues showed that mice with C-terminal 36-amino acid-deleted Hsp90α, theoretically preventing the protein to form a dimer, had little phenotypic difference from their wild-type counterparts. However, the lack of the Hsp90α chaperone function, even in the presence of a higher level of Hsp90β, specifically paralyzed the production of sperm in male mice [11]. Interestingly, an earlier study by Yue and colleagues reported that a reduced level of Hsp82 (*E(sev)3A*) was associated with a specific defect in spermatogenesis in *Drosophila* [21]. In addition to a defect in spermatogenesis, Imai and colleagues reported that the destruction of Hsp90α’s chaperone function caused defect in extracellular antigen translocation across the endosomal membrane into the cytosol in both male and female mice [12]. A recent study provided a likely explanation for why Hsp90α is specifically involved in spermatogenesis. While most tissues keep oxygen pressure between 2% and 9% (the oxygen levels in mouse circulation), the testis is known to have a constant oxygen pressure lower than 1.5%, as well as a temperature two degree lower than the rest of the body [33]. Accordingly, the low oxygen pressure causes a constitutive expression of hypoxia-inducible factor-1alpha (HIF-1α) in the premeiotic cells of the mouse testis [34,35] and in human sperm [36]. Tang et al. showed that CRISPR-cas9-mediated knockout of Hsp90α destabilized HIF-1α in mouse testis, resulting in infertility in male mice [13]. More interestingly, the requirement of Hsp90α for supporting spermatogenesis has been shown during the adult life of mice as well [37]. We have recently found that the testis is the only organ in mice that constitutively expresses a higher level of HIF-1α, providing a mechanism for why Hsp90α knockout specifically affects the testis’ function. We speculate that, beside the specific role of Hsp90α in stabilizing HIF-1 during spermatogenesis, the evolutionarily conserved duty for Hsp90α, both intracellularly and extracellularly, is to deal with stress-related pathophysiological conditions, such as wound healing, tissue inflammation and tumor progression. In contrast, Hsp90β serves as the critical housekeeper inside the cells.

## 6. The Chaperoning Data In Vitro Do Not Match the Biology In Vivo between Hsp90α and Hsp90β

Even though the role for Hsp90α and Hsp90β during mouse development dramatically varies, with Hsp90α knockout only affecting spermatogenesis and Hsp90β knockout causing death, similar degrees of difference at the molecular level, such as the profiles of their client proteins and co-chaperones, have never been established. In contrast, the results of the limited studies that compared the profiles of Hsp90α- vs. Hsp90β-associated signaling molecules have consistently shown much fewer variations than similarities. Taking the most critical PI-3K—Akt pathway that regulates cell metabolism and the EGFR-Ras-Raf-MEK-ERK signaling pathway that controls cell growth, Taherian et al. showed that Hsp90α and Hsp90β equally bind to Raf-1 and MEK1 under physiological conditions [27]. Tang and colleagues knocked out Hsp90α or knocked down Hsp90β or both in MDA-MB-231 cells and measured the steady-state stability of EGFR, Akt1, Akt2, Erk1/2 and cyclin D1. Their results show that the absence of either Hsp90α or Hsp90β alone did not cause any significant degradation of these signaling molecules, except that the absence of Hsp90β affected the stability of Akt2, Erk1/2 and cyclin D1 more than Hsp90α absence, whereas the absence of both Hsp90α and Hsp90β caused a catastrophic degradation of all the signaling molecules of the pathway [38]. If these results were interpretated as Hsp90α and Hsp90β compensating for each other’s absence, it would not explain the dramatically different outcomes of Hsp90α knockout and Hsp90β knockout in mice. If the interpretation were instead that Hsp90β, but not Hsp90α, chaperones a yet unidentified factor independent of the EGFR-Ras-Raf-MEK-ERK signaling pathway and critical for mouse development, it would be of a great interest to identify such a factor(s). 

## 7. A Main Hurdle for Hsp90-Inhibiting Therapeutic Development: The Biomarker Selection

Since 1999, the intrinsic ATPase of heat shock protein-90 (Hsp90) family proteins has been the target of several generations of structurally improved small-molecule inhibitors in more than 60 cancer clinical trials [1,2,3,4,5]. The initial excitement came from the realization that Hsp90 acts as a chaperone to stabilize almost all key signaling molecules in parallel signaling pathways and, therefore, the inhibition of Hsp90 may turn out to have a one-fell-swoop effect, both blocking tumor progression and preventing drug resistance. Major scientific support behind the clinical trials came from a 1994 landmark study showing that 17-AAG destabilizes the oncogenic v-src tyrosine kinase by directly binding and dissociating Hsp90 from chaperoning the v-src kinase in v-src-transformed mouse NIH3T3 fibroblasts and PC3 human prostate cancer cell lines [39]. To date, however, few inhibitors of this kind have received FDA approval for the treatment of cancers in humans. Most experts attributed the spectacular failures to various, otherwise correctable, factors and remained optimistic for the ultimate success of the approach, despite the evaluations largely being speculative [2,4,6], such as the pan-inhibition of all the four cytosolic Hsp90-related isoforms, including the two organelle-residing isoforms, Grp94 and TRAP1. However, some cautioned that Hsp90 may not be a viable anti-cancer target to begin with [3].

The online FDA’s drug development guidelines list the step-by-step approach: (1) discovery and development, i.e., the identification of a drug candidate; (2) preclinical in vitro and in vivo research; (3) clinical evaluations; (4) FDA drug review and (5) FDA post-market drug safety monitoring (https://www.fda.gov/patients/drug-development-process/step-2-preclinical-research, accessed on 15 November 2022). However, even prior to the first 17-AAG (17-*N*-allylamino-17-demethoxygeldanamycin) clinical trial in 1999, few preclinical studies that provide evidence for a druggable window between cancer and normal cells for the inhibitor could be found from public domains. Only after clinical trials proceeded anyway, a number of publications reported higher accumulation or the higher sensitivity of tumor cells or tumor-bearing mice to the ATP-binding inhibitors than normal cells or control animals. Kamal et al. reported a 100-fold difference in binding affinity of the cell-free Hsp90 protein complex from tumor cells versus normal cells to 17-AAG [40]. Similar in vitro binding results were reported for purine-based Hsp90 inhibitors [41,42,43,44]. Vilenchik and colleagues reported that normal cells are 10- to 50-fold more resistant to PU24FCI’s anti-proliferation effects than tumor cells [45]. The increased binding affinity in tumor cells was thought to be due to an increased level of post-translational modifications [40,44,45]. Furthermore, Solit et al. showed that the maximally tolerated dose of 17-AAG was higher in control mice than in tumor-bearing mice [41]. In contrast, several studies of 17-AAG and 17-DMAG (17-Dimethylaminoethylamino-17-demethoxygeldanamycin) in intact cells showed rather conflicting results [46,47,48]. In contrast to the above cell-free protein binding results, data using intact cells in response to 17-AAG, 17-DMAG (17-Dimethylaminoethylamino-17-demethoxygeldanamycin) and purine-based inhibitors were much less effective and unclear. For instance, Premkumar et al. reported a moderate difference in cellular toxicity, between 20% in normal cells and 50% in cancer cells, to 17-AAG [47]. Similarly, Lukasiewicz and colleagues showed 30–50% normal cell death versus 55–80% cancer cell death under treatment with 17-AAG or 17-DMAG [48]. Instead of a thousand-fold difference in cell-free binding assays, two groups showed a modest 20-fold difference in cell growth inhibition between tumor cells versus a normal cell line [42,43]. Furthermore, Vilenchik et al. reported a less than 10-fold difference in the IC50 values for inhibitor PU24FCI in cell growth inhibition between two normal and 15 cancer cell lines [45]. Close to two dozen ATP binding inhibitors of Hsp90 proteins with five distinct scaffolds have since been approved for cancer clinical trials in the past two decades [3,4]. The common drug toxicity biomarker used in the clinical trials was peripheral blood mononuclear cells (PBMCs), obviously for reasons of convenience, since it assumes that all other organs, tissues and cell types in the human body share an identical response as PBMCs to the inhibitors. 

A recent study by Tang and colleagues pointed out a possible critical defect in the design of previous clinical trials: the limited and biased biomarker selections. They showed that, while Hsp90α and Hsp90β make up the threshold of the total chaperoning capacity in the cells, the levels of Hsp90 expression dramatically vary, especially among non-transformed cells and mouse organs. More importantly, the highly variable levels of Hsp90, from as low as 1.7% to as high as 9% of the total cellular proteins in non-cancer cells, directly correlated with either extreme sensitivity or extreme resistance to a classical Hsp90 ATP-binding inhibitor [38]. In mouse organs, as previously shown in Figure 1C, the relative expression of Hsp90β remains compatible among different mouse organs, whereas the relative expression of Hsp90α varies dramatically. Therefore, the detected variations in sensitivity and resistance among different cell lines were likely due to the variable titration effects of Hsp90α. Similarly, the same client proteins from the mitogenic pathway in cells exhibited unexpected and heterogenous reactions among eight randomly selected cancer cell lines in response to the Hsp90 ATP-binding inhibitor, inconsistent with the classical understanding. These findings of Tang and colleagues could complicate patient selection, biomarker choices, toxicity readout and clinical efficacy for Hsp90-ATP-binding inhibitors [38]. Specifically, this study sends several clinically relevant messages to currently ongoing and future cancer clinical trials with Hsp90 ATP-binding inhibitors. First, the toxicity-tolerable dosages may vary among different types of normal cells, tissues and organs in human patients, which would be technically difficult to monitor across them all. As schematically shown in Figure 3, the predicted expression of Hsp90 could vary dramatically in different human organs, based on the data of mice [11,12,13]. Any potential misinterpretations of some “overall toxicity” data from the PBMC cells alone could potentially cause unexpected post-clinical trial damage in patients. Second, there has been little evidence that the same type of tumor in different patients shares a similar Hsp90 level, posing the difficulty of choosing the right dosage for treatment, especially during late stages of clinical trials. It may be necessary to measure the tumor Hsp90 level in each individual cancer patient, group the patients with similar Hsp90 levels and treat different groups of cancer patients with different dosages of the inhibitor in clinical trials. If this concern proves true, it greatly complicates the design of a cancer clinical trial. In retrospect, the wide range of Hsp90 levels in normal tissues and in the same tumor in different patients could have in fact caused the failures of previous clinical trials. 

## 8. Choice of Hsp90α, but Not Hsp90β, for Extracellular Functions by Evolution

Extracellular Hsp90, especially reported as “cell surface-bound” Hsp90, was reported as early as the late 1970s by a number of laboratories for its role in neuronal differentiation, autoimmune disease and tumorigenesis in mouse and human cultured cells [49]. Those early studies did not and could not clearly distinguish whether it was Hsp90α or Hsp90β involved in each of those processes. Not until the 2020s, two laboratories were searching for a secreted protein(s) that supports tumor cell invasion and skin cell migration during wound healing, were two distinct and mechanistically related pathophysiological processes identified. Jay’s group at Tufts University first reported that a secreted protein from the conditioned medium of a fibrosarcoma cell line, HT-1080, promoted tumor cell invasion in vitro by activating the matrix metalloproteinase 2 (MMP2) [50]. Li’s laboratory at the University of Southern California reported that the purification of a secreted protein from the conditioned medium of hypoxia-stressed primary human dermal fibroblasts and keratinocytes strongly stimulated skin cell migration in vitro and promoted wound healing in mice [51,52]. The common protein involved in both tumor cell invasion in vitro and wound healing in vivo was the secreted form of Hsp90α, collectively called either extracellular Hsp90α or eHsp90α, which includes cell-surface-bound, extracellular-vesicle (EV)-bound and non-EV-bound Hsp90α. While the percentage of the “cell surface-bound” Hsp90 has never been calculated, a recent study found that the ratio of the secreted eHsp90α via the endosomal/exosomal pathway and the autophagosomal pathway (both belong to the so-called type III secretion pathway [53] is 1:9. Since then, except less than a hand full studies on eHsp90β, the vast majority of studies have identified the specific involvement of eHsp90α. In the extracellular environment, eHsp90α offers two main biological functions, protecting cells from stress-triggered apoptosis and promoting cell migration (but not proliferation), both of which are crucial during tissue repair and tumor progression [49]. A variety of mechanisms have been reported to execute these two main functions, including eHsp90α as a ligand that binds to the cell surface receptor, LRP-1, and activates the Akt kinases [54,55], regulating MMP2 activity [56] via a TIMP2-AHA1 switch [57] or manipulating ECM [58]. Zou and colleagues identified that a dual lysine motif distinguishes Hsp90α from Hsp90β for extracellular functions. The lysine-270 and lysine-277 motif in human Hsp90α is conserved in all Hsp90α subfamily members from fish to humans (see Figure 2A). However, the two corresponding amino acid residues in human Hsp90β and other Hsp90β subfamily members, from chickens to humans, are replaced by glycine and threonine (see Figure 2B). Zou et al. demonstrated that swapping these two amino acids between human Hsp90α and human Hsp90β completely eliminated the extracellular functions of eHsp90α in vitro and in vivo and, in reverse, it granted Hsp90β the eHsp90α-like pro-motility activity in vitro, but not wound healing activity in vivo [9]. These findings suggest the dual lysine motif is necessary, but not sufficient, to convert eHsp90β to an eHsp90α-like molecule to support wound healing and tumorigenesis in vivo, and an additional element(s) is likely required to complete these more complex missions in vivo. 

## 9. Why Does Metastasis of the Same Tumor Occur in Some Patients but Not Others?

The five recognized common steps of cancer development include gene mutations, hyperplasia, dysplasia, primary tumor formation and tumor metastasis [59,60,61,62]. While most cancer patients die from cancer cell metastasis, the majority of the United States Food and Drug Administration (FDA)-approved oncology drugs (>1000 by end of 2021) target the primary tumors [63]. These drugs extend patients’ survival for variable periods of time, but gradually lose efficacy following as short as several months of treatment due to new mutations in the tumors. Accordingly, the cost of developing this kind of anti-cancer therapeutics has not always been justifiable from a business perspective. In theory, unlike drugs targeting primary tumors, drugs targeting the metastasis-promoting factors could be broadly beneficial, since these factors are likely shared among different cancers but unlikely subject to mutations in a cancer-specific fashion. There is compelling evidence that the yet unidentified “metastasis-promoting factors” belong to stress response factors involved in tissue injury, inflammation, wound healing and angiogenesis in the host, but taken advantage of by tumors [59,62]. McAllister and Weinberg argued that cancer is a systemic disease, in which successful cancer metastasis is not only determined by the primary tumor and its immediate microenvironment (TME), but also by a patient’s other pre-existing pathological conditions [64]. For instance, Roy and colleagues reported that a significant increase in breast cancer-associated secondary metastasis to the lung and bone was observed in arthritic versus non-arthritic PyV MT mice, which correlated with increased pro-inflammatory cytokines [65]. Demicheli et al. reported that tumor recurrence dynamics does not depend on the site of metastasis; instead, the timing of recurrences is generated by factors influencing the metastatic development regardless of the seeded organ [66]. Thus, surgery to remove the primary tumor often terminates tumor dormancy, resulting in accelerated relapses [67]. Furthermore, Gupta et al. showed that, despite lacking ER expression, tumors arising following pregnancy required circulating estrogens for growth. Increasing the circulating estrogens promoted the formation and progression of ER-negative cancers, suggesting a novel mechanism by which estrogens promote the growth of ER-negative cancers [68]. In summary, the findings of these studies suggest that two combined chronic pathological conditions in the same cancer patients increase the success rate of tumor metastasis. The molecular basis for the chronic conditions is tissue inflammation, which could influence the favorability of the niche toward tumors. Indeed, patients with chronic inflammatory conditions are at a higher risk for cancer-associated mortality [69]. In addition, studies in the lung have shown that acute inflammatory responses concurrent with the experimental administration of tumor cells in the blood can promote metastatic development [70]. We speculate that eHsp90α could be a candidate for such a stress-responding factor involved in tumor metastasis. Tens of thousands of patients with all NIH-listed major types of cancers, including breast, lung, stomach, colon, liver, uterus, prostate, brain, pancreas, esophagus, blood, head and neck, and skin, have been shown to exhibit dramatically elevated plasma eHsp90α up to 10-fold over their healthy controls. The plasma Hsp90α levels closely correlate with the early-to-late stages of the cancers [71,72,73,74,75,76,77,78,79,80,81,82,83,84,85,86,87]. Likewise, patients with idiopathic pulmonary fibrosis (IPF), Crohn’s disease, psoriasis, chronic glomerulonephritis, amyotrophic lateral sclerosis, obesity, systemic sclerosis and diabetes, showed significantly increased plasma eHsp90α [87,88,89,90,91,92]. As illustrated in Figure 4, for instance, IPF-caused chronic inflammation could increase tumor metastasis to the lung. It is of great interest to investigate the hypothesis of cancer being “a systemic disease” and the participation of other chronic disorders in (successful) cancer metastasis, specifically, whether eHsp90α in the circulation of cancer patients with other chronic pathological conditions adds a critical contributor for cancer metastasis [49]. 

## 10. What Is the Therapeutic Window to Target Hsp90 in Cancer Next?

Based on the above findings and analyses of Hsp90α or Hsp90β, we would like to make two recommendations for future anti-Hsp90 therapeutic development. First, there is a need to develop Hsp90β-specific inhibitors [93,94]. Hsp90β is the critical chaperone for life and its expression remains constant in different organs, at least based on the data in mice. Since the new generation of Hsp90β-specific inhibitors will not be titrated by the variable levels of Hsp90α in different cells and organs, the toxicity profiles in different patients should be compatible. In addition, it may be necessary to screen and choose the patients whose tumor biopsies show a higher Hsp90β expression than normal organs for clinical trials. Second, increasing lines of evidence have indicated that eHsp90α plays an essential role in primary tumor invasion, as well as subsequent metastasis. Studies of the next few years will use animal models to investigate if the elevated plasma eHsp90α in human cancer patients increases the success of tumor metastasis. We recommend neutralization of the elevated eHsp90α in the circulation by monoclonal antibodies [9,71,95]. The obvious advantages of targeting plasma eHsp90α include: (1) it being easier to access without penetrating tissue barriers and (2) it being less toxic, as eHsp90α is only a stress-responding factor. In fact, eHsp90α could well be the actual target of the recently approved oral tablet, the Jeselhy^®^ Tablets 40 mg (pimitespib, TAS-116) in Japan, around the gastrointestinal stromal tumor (GIST) in the intestine, bypassing the potential toxicity of intravenous injection. 

## 11. Perspective

Targeting Hsp90 in cancer has been a long and frustrating journey, despite initial excitement and expectations. It reminds us all that each of the NIH-guided sequential steps of drug development must be thoroughly followed, fully completed and independently evaluated, prior to entering clinical trials. The good news is that we have gained critical new insights into the core of the problem over the past decade. In our opinion, Hsp90, such as Hsp90β and eHsp90α, will ultimately prevail as viable targets of anti-cancer drugs that will not allow the cancers to develop resistance. 

## Figures and Tables

**Figure 1 cells-12-00277-f001:**
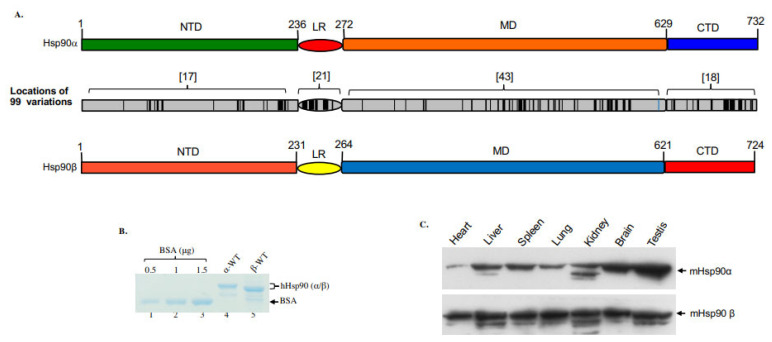
Variations in location, molecular mass and expression in mouse organs between Hsp90α and Hsp90β. (**A**) Locations (black colored) of the 99 amino acid substitutions and deletions between human Hsp90α and Hsp90β, in which the LR shows the highest percentage of the mutations. (**B**) The molecular mass of human recombinant Hsp90α and Hsp90β on an SDS-PAGE gel. (**C**) The relative expression of Hsp90α and Hsp90β in indicated mouse organs in reference to equalized total proteins of the various organs.

**Figure 2 cells-12-00277-f002:**
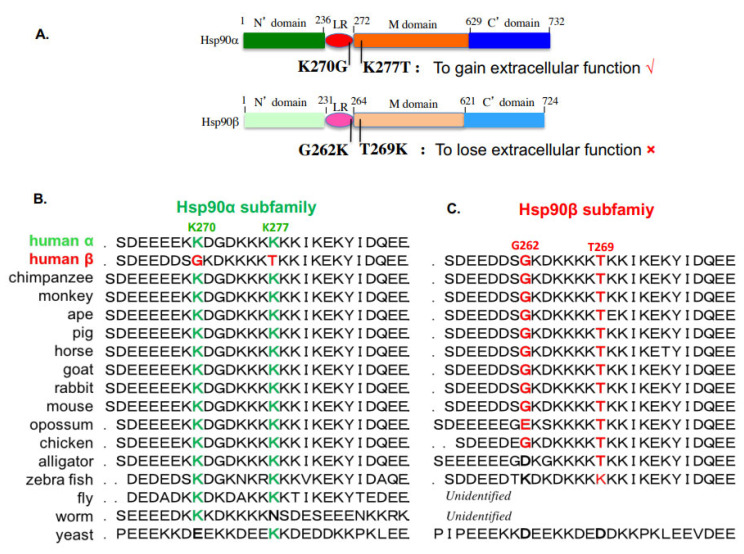
Evolutionarily conserved lys-270 and lys-277 distinguish Hsp90α from Hsp90β for having important extracellular functions. (**A**) The dual lysine motif is essential for the extracellular and ATPase-independent functions of eHsp90α. Substitutions of G262 and T269 in Hsp90β with lysine grant Hsp90β similar extracellular functions to eHsp90α. The dual K270/K277 motif in human Hsp90α (**B**) and the corresponding G262/T260 motif in human Hsp90β (**C**) are evolutionarily conserved in all the Hsp90α subfamily and the Hsp90β subfamily members, respectively.

**Figure 3 cells-12-00277-f003:**
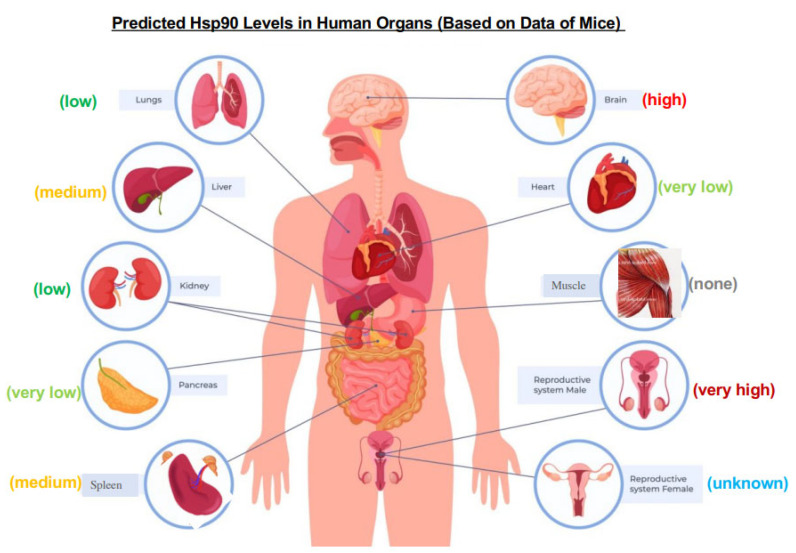
The relative Hsp90 levels in different organs based on studies in mice. Assuming that the profiles of Hsp90 expression in different organs of mice and humans are compatible, the relative levels in various human organs are predicted based on anti-(pan) Hsp90 antibody western immunoblotting data of mouse organs. (1) none: undetectable; (2) very low: weakly detectable; (3) low: visible; (4) medium: standard intensity; (5) high: strong intensity; and (6) very high: overexposed intensity. The image of human anatomy was taken and modified from Human Health UK (https://homehealth-uk.com/malebodydiagram/, accessed on 1 December 2022).

**Figure 4 cells-12-00277-f004:**
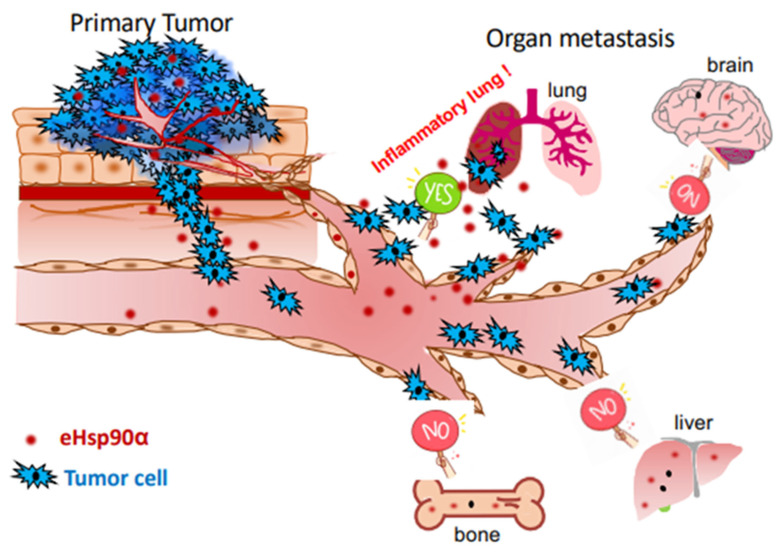
A hypothetical illustration of tumor metastasis being a systemic disease involving tissue inflammation-triggered secretion of eHsp90α. A primary tumor from the organ of its origin is unable to succeed on metastasis without chronic damage of a distant organ, such as IPF lung. The chronic inflammation of the damaged organ attracts and welcomes circulating tumor cells (CTCs) to their new home. Participation of tissue damage-triggered secretion of eHsp90α in tumor metastasis has been supported by recent studies.

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
