# Peer review of "The Distinct Assignments for Hsp90α and Hsp90β: More Than Skin Deep"

_cells, 2023, doi:10.3390/cells12020277_

Round 1

Reviewer 1 Report

The review written by Chang et al. provides an overview of Hsp90 story, distinguishing a role of Hsp90a and Hsp90b. It covers not only ontogeny, mouse and tissue development, but also cancer metastasis, and the therapeutic perspective. The content is mostly updated, well-balanced, and is full of deep insight. Only a point concerning a definition of extracellular Hsp90 might have a confusion with Hsp90 contained within extracellular vesicles such as exosome produced by cancer cells and/or normal cells. This point might need to clarify at some points, if possible.

Overall, I think this review article should be published as soon as possible.

Author Response

Dear Reviewer #1:

An article cannot be complete and significant without the diligent review like yours. We thank you for your time, encouragement and suggestions. We have revised the article based on your valuable comments (see red inked corrections).

Sincerely yours,

Wei Li

Reviewer 2 Report

The manuscript of Cheng Chang et al. is dedicated to a discussion of the possibilities for using HSP90 family proteins as targets for anticancer therapy. It is known that tumor cells are characterized by an increased level of expression of protective heat shock proteins, in particular HSP90; therefore, one of the currently developed approaches to antitumor therapy is aimed at suppressing the expression and functional activity of HSP90. However, these approaches have not yet demonstrated the expected efficacy, both in preclinical and clinical trials. The authors of this literature review conducted a detailed analysis of the possible reasons for the low efficacy of the HSP90 inhibitors used. The review focuses on the analysis of differences in the role and functional activity of two varieties of these proteins, HSP90 alpha and HSP90 beta on different levels and in different organisms. This emphasis determines the novelty and originality of the submitted manuscript. The authors of this review consider in detail the cytosolic Hsp90 isoforms, Hsp90 alpha and Hsp90 beta in lower organisms and in mammalian cells, Hsp90 alpha and Hsp90 beta during mouse development, and much more. Based on the performed analysis, the authors described the main obstacles that make it difficult to obtain a significant antitumor effect when using the developed drugs for HSP90 targeted therapy. In the final part of the article two recommendations for future anti-HSP90 therapeutic development of such drugs were proposed.

The significance of this review is beyond doubt, since the presented summary of literature data and their analysis give an objective integral picture of existing approaches and ideas on the indicated topic and highlight possible ways of practical use of the accumulated knowledge.

In general the review is well presented; the data are of considerable novelty and interest. However, the manuscript contains a number of inaccuracies and misprints in the text. For example page 9:

   members from xxx to humans (see Figure 3A). However, the two corresponding amino acid residues in human Hsp90 beta and other Hsp90 beta subfamily members from xxx to humans are replaced by glycine and threonine (see Figure 3B)    “.

The authors should correct the mistakes.

Author Response

Dear Review #2:

An article cannot be complete and significant without the diligent review like yours. We thank you for your time, encouragement and suggestions. We have revised the article based on your valuable comments (see blue inked corrections).

Sincerely yours,

Wei Li
